SciPost Physics

Submission

# Density of states correlations in Lévy Rosenzweig-Porter model via supersymmetry approach

Elizaveta Safonova[1,2], Aleksey Lunkin[1] and Mikhail Feigel' man[1,3]

**1** Nanocenter CENN, Ljubljana, Slovenia
**2** Department of Physics, University of Ljubljana, Slovenia
**3** Jozef Stefan Institute, Ljubljana, Slovenia

June 7, 2025

## Abstract

We studied global density-of-states correlation function $R(\omega)$ for **Lévy-Rosenzweig-Porter** random matrix ensemble [1] in the non-ergodic extended phase. Using extension of Efetov's supersymmetry approach [2] we calculated $R(\omega)$ exactly in all relevant ranges of $\omega$. At relatively low $\omega \leq \Gamma$ (with $\Gamma \gg \Delta$ being effective mini-band width) we found GUE-type oscillations with period of level spacing $\Delta$, decaying exponentially at the Thouless energy scale $E_{Th} = \sqrt{\Delta\Gamma/2\pi}$. At high energies $\omega \gg E_{Th}$ our results coincide with those obtained in Ref. [3] via cavity equation approach. Inverse of the effective mini-band width $1/\Gamma$ is shown to be given by the average of the local decay times over Lévy distribution.

## 1  Introduction

There are numerous indications for the apparent absence of thermalization and breakdown of ergodicity in large interacting quantum systems [4, 5, 6] with sufficiently high degree of disorder. However, almost no exact theoretical results are available, making reliable interpretation of real and numerical experiments rather complicated. While original theoretical approach to this problem [7, 8] was concentrated on low-temperature transport properties, later development in this field (called now Many Body Localization (MBL) problem) was shifted mainly to the infinite-temperature limit, for the sake of simplification; also, some types of experiments (NMR, cold atoms) may indeed be realized at the effective temperatures much above typical energies involved in the Hamiltonian. Still the issue of existence of non-ergodic and/or MBL state in a real physical system with short-range interaction is highly debatable [9, 10].

One of major obstacles for the theory of MBL phenomena is the presence of well-developed spatial correlations. Indeed, while dimension of Hilbert space of a random system containing $n$ spins-$\frac{1}{2}$ is $2^n$, the number of parameters entering its Hamiltonian is just $\sim n^2$ at most. Proper account of these correlations is not developed yet, and theoretical results are limited to some artificial models where these correlations are absent. In particular, it was shown in Ref. [11] that structure-less Quantum Random Energy Model possesses three different phases, depending on macroscopic energy and degree of disorder: ergodic, fully localized (MBL) and intermediate non-ergodic extended (NEE) state. Theoretical demonstrations of these features were obtained by means of approximate mapping of the QREM Hamiltonian to the Rosenzveig-Porter matrix model shown previously [12] to have all three such phases. It was understood later on [13, 1] that Gaussian RP model [12] is too simplified to describe more realistic problems; one possible way to generalize this model is to account for the possibility of fat-tail distribution of non-diagonal matrix elements. An independent reason to be interested in this kind of models is due to (numerical) observations of a power-law distribution of matrix elements connecting different bit-strings in a systems of interacting quantum spins [14, 15, 16].

Invariant Lévy matrix ensemble was introduced originally in Ref. [17] and its Rosenzveig-Porter version was studied in Ref. [1, 3]. In particular, Ref. [1] demonstrated the presence of NEE state in the whole range of parameters $\mu, \gamma$ characterizing the model, while in Ref.[3] full

description of local density-of states correlations at large energy difference (effectively, setting level spacing to zero) were obtained by means of statistical analysis of cavity equations. However, to study level correlations at not-so-large energies a more elaborated technique is needed. Well-developed methods to treat this type of problem in usual random-matrix ensembles are based on the super-symmetry method due to Efetov [18]. Application of this method to Gaussian RP model was recently provided in Ref. [19]. However, standard SUSY method based upon Hubbard-Stratonovich transformation of the functional integral is not appropriate for matrix models with a heavy-tail distributions, especially when second moment of the distribution diverges, as in the Lévy case. More general approach to the construction of super-symmetric field theory for disordered quantum systems was proposed in Ref. [2], where functional generalization of the Hubbard-Stratonovich transformation was introduced. In the previous paper [20] we employed this approach to study average density of states of Lévy-Rosenzveig-Porter ensemble. Below we extend this approach for the calculation of the global density-of-state correlation function $R(\omega) = \langle \rho(E + \omega/2)\rho(E - \omega/2)\rangle/\langle \rho(E)\rangle^2$ at arbitrary $\omega$ in the NEE state. We demonstrate the presence of three energy scales in the problem: mean level spacing $\Delta$, typical mini-band width $\Gamma \gg \Delta$ and intermediate scale $E_{Th} = \sqrt{\Gamma\Delta/2\pi}$ which plays the role of a Thouless energy in our problem. Previous results [3] are confirmed for $\omega \gg E_{Th}$ by our super-symmetry method, while at low $\omega \leq E_{Th}$ the function $R(\omega)$ demonstrate oscillations typical for Wigner-Dyson random matrix ensembles.

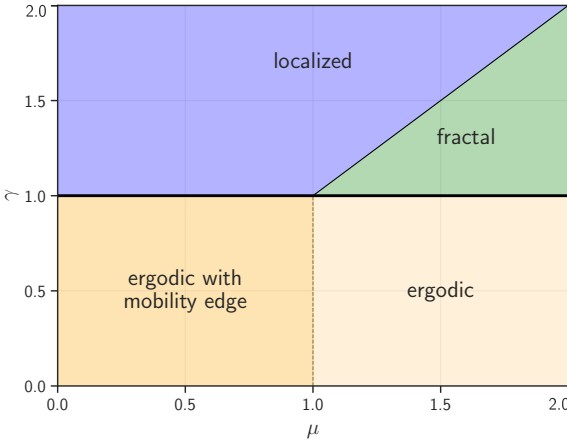

Figure 1: Different regimes depending on the width of mini-band $\Gamma_0$ in comparison with level spacing. $\Gamma_0$ depends on the system size $N$ and the parameters $\gamma, \mu$ as $\sim N^{\frac{1-\gamma}{\mu-1}}$ and determines behavior of the system (See (22) and comment under it). At $\gamma < 1$ we see ergodic states; if $\mu < 1$, then there is also mobility edge (transition to localized states at energies closer to the band edge). We are interested in the range of $\gamma > 1$ where eigenstates are either localized ($\gamma > \mu - 1$) or extended but non-ergodic at $\gamma < \mu - 1$. In this latter phase $\Gamma_0$ is much larger than level spacing but much smaller than with whole band width $W$.

Before going into the calculations, we briefly review the main features of the phase diagram for Lévy-RP matrices, based mostly on Ref.[21]. The part of the phase diagram we're interested in covers the range $1 < \mu < 2$, and it's shown in Fig.(1). The different phases are defined based on the behavior of the eigenvectors $\psi_n(i)$. These can be ergodic, where the inverse participation ratio (IPR) scales like $I(N) \sim N^{-1}$, localized with $I(N) \sim$ constant, or non-ergodic but extended — with $I(N) \sim N^{-D}$ for some $0 < D < 1$. There are two

ergodic (E) phases. One appears for $1 < \mu < 2$, where all eigenvectors are ergodic for any energy $E_n$. The other is for $0 < \mu < 1$, where a mobility edge $E_0$ separates ergodic and localized states: eigenvectors are ergodic when $|E_n| < E_0$ and localized when $|E_n| > E_0$. All three phases—ergodic, localized, and non-ergodic extended—meet at the tricritical point $\mu = \gamma = 1$.

In this paper we are concerned with the correlation function defined by Eq.(7) in at $1 < \mu < 2$. We show that Levy-RP model indeed experiences phase transitions at the boundaries $\gamma = \mu$ and $\gamma = 1$ and we provide the explicit analytical calculation.

The rest of the paper is organized as follows. Sec.2 introduces definitions of the random matrix ensemble we are going to study and representation of the correlation function $R(\omega)$ in terms of functional integral over super-fields. Sec. 3 describes functional Hubbard-Stratonovich transformation and provides saddle-point analysis of the relevant functional integral. Sec. 4 is devoted to calculation of the correlation function $R(\omega)$ in two overlapping limiting cases: high frequencies $\omega \gg E_{Th}$ and low frequencies $\omega \ll \Gamma$. Since $E_{Th} \ll \Gamma$, we thus obtain full behavior of $R(\omega)$ in the whole range of frequencies. Sec. 5 contains our conclusions. Supplemental material (Secs. A-F) contains technical details of our calculations.

## 2 Definitions

### 2.1 The matrix ensemble

Our research object is an ensemble of $N \times N$ complex hermitian matrix $\hat{H}$ which can be represented as the sum of two matrices:

$$\hat{H} = \hat{H}_D + \hat{H}_L, \tag{1}$$

where $\hat{H}_D$ is a diagonal random matrix with real independent and identically distributed (i.i.d.) entries and $\hat{H}_L$ is a full matrix where *all* elements are i.i.d. The distributions of $\hat{H}_L$ and $\hat{H}_D$ are generally different. We consider the case of the *L'evy-Rosenzweig-Porter (L'evy-RP) matrices* [1] where the entries of $\hat{H}_D$ are random, broadly distributed with the typical distribution width $W$ so that $W$ is the largest energy scale. Level spacing $\Delta \sim W/N$ is the smallest energy scale. $H_L$ entrees are complex and defined as follows:

$$[H_L]_{mn} = h_{mn} \exp\left(i\theta_{mn}\right), \quad h_{nm} \geq 0, \quad -\pi \leq \theta_{mn} < \pi. \tag{2}$$

The phase $\theta_{nm}$ distributed uniformly with $P_\theta\left(\theta\right) = \frac{\theta(\pi - |\theta|)}{2\pi}$ and the amplitudes $h_{mn}$ have a distribution according to the power law. For convenience, we chose the particular one-sided L'evy distribution

$$P_L^{(\mu,\gamma)}\left(h_{mn}^2\right) = \frac{N^{\frac{2\gamma}{\mu}}}{\sigma^{2/\mu}} L_{\mu/2}\left(\frac{N^{\frac{2\gamma}{\mu}}}{\sigma^{2/\mu}} h_{mn}^2\right), \tag{3}$$

where $\sigma$ is an energy unit and $L_{\mu/2}(x)$ is *one-sided* Lévy stable distribution [22, 23] which is defined by Laplace characteristic function:

$$\tilde{L}_{\mu/2}(r) \equiv \int_0^\infty L_{\mu/2}(x) e^{-rx} dx \equiv e^{-r^{\mu/2}}, \quad 1 < \mu \leq 2. \tag{4}$$

We chose that particular function because it supports only positive values and has a convenient representation in terms of its Laplace transform. Using Eqs.(4),(3) we can find the Laplace characteristic function of rescaled $P_L$ distribution:

$$\int_0^\infty P_L\left(h^2\right) e^{-rh^2} d\left[h^2\right] \equiv \exp\left(-\frac{\sigma}{N^\gamma} r^{\mu/2}\right), \qquad \begin{matrix} 1 < \mu < 2 \\ \gamma > 0 \end{matrix} \tag{5}$$

In fact, any distribution with the same power law tail will lead to similar results, as explained in the end of the paper. Function (3) has the following power law asymptotics

$$P_L\left(h^2\right) dh^2 \approx \frac{\mu\sigma^\mu dh}{\Gamma\left(1 - \frac{\mu}{2}\right) N^\gamma h^{1+\mu}}, \qquad 1 < \mu < 2 \tag{6}$$

For $\mu \geq 2$ this distribution has a finite variance and the model becomes equivalent to the usual Gaussian Rosenzweig-Porter model. To compare intermediate results with the previous papers [19], [21], one can put $\sigma = 1$, while notations of Ref. [3] are recovered by the choice $\gamma = 1$ and $\frac{\sigma^\mu}{\Gamma\left(1 - \frac{\mu}{2}\right)} = h_0^\mu$.

Notice that while the variance $W^2$ of $H_D$ is independent of the matrix size $N$, the typical value of $H_L$ scales with $N$ as $\sigma N^{-\gamma/\mu}$, and its variance diverges at $\mu < 2$ due to the tail in $L_{\frac{\mu}{2}}(x^2) \sim x^{-(1+\mu)}$. There is a special value $\mu = 2$ where the distribution $L_{\mu/2}(x)$ reduces to the delta function $\delta(x - 1)$.

## 2.2   Global DoS correlation function and supersymmetric method

Our goal is to calculate correlation function of global density of states which is defined as

$$R\left(E, \omega\right) = \frac{\left\langle \rho\left(E + \frac{\omega}{2}\right) \rho\left(E - \frac{\omega}{2}\right)\right\rangle}{\left\langle \rho\left(E\right)\right\rangle^2} \tag{7}$$

where $\rho(E) = -\frac{1}{\pi N}\mathrm{Tr}\,\mathrm{Im}\,\hat{G}_R(E)$ is density of states (DoS) and $\hat{G}_R(E)$ is retarded Green function of the Hamiltonian (1) at energy $E$. It is convenient to choose the scaling so that DoS becomes a function of the order of unity:

$$\frac{1}{\Delta} = N\rho\left(E\right), \qquad \int dE\rho\left(E\right) = 1 \tag{8}$$

where $\Delta$ is mean level spacing. To continue the calculation one should switch to the Green function representation, so that the correlation function is

$$R\left(E, \omega\right) = \frac{1}{2} + \frac{\Delta^2}{2\pi^2}\mathrm{Re}\left\langle \mathrm{Tr}\hat{G}_R\left(E + \frac{\omega}{2}\right) \mathrm{Tr}\hat{G}_A\left(E - \frac{\omega}{2}\right)\right\rangle \tag{9}$$

Two-point correlation function can be expressed through differentiation the partition function $Z(E, \omega, J_A, J_R)$ over background fields $J_R, J_A$. The partition function $Z(E, \omega, J_A, J_R)$ is given by the integral over super-vectors $\psi_i$ (for the derivation, see Supplement, Sec.A.1).

$$R\left(E, \omega\right) = \frac{1}{2} + \frac{\Delta^2}{8\pi^2}\mathrm{Re}\frac{\partial^2 Z\left(E, \omega, \hat{J}\right)}{\partial J_R \partial J_A}\Bigg|_{J_{R,A}=0} \tag{10}$$

$$Z\left(E, \omega, \hat{J}\right) = \left\langle \int [d\psi]\exp\left(i\sum_{n,m}\psi_n^\dagger \hat{L}\left(\left[E + \frac{\Omega}{2}\hat{L} - \hat{J}\hat{K}\right]\delta_{nm} - H_{nm}\right)\psi_m\right)\right\rangle_{\hat{H}} \tag{11}$$

where $\Omega \equiv \omega + i0$ (here and below infinitesimal imaginary is introduces to guarantee convergence of the integrals). Expression (11) uses super-algebra formalism which includes commuting and anti-commuting variables:

$$\psi_i = \begin{pmatrix} \psi_R \\ \psi_A \end{pmatrix} = \begin{pmatrix} S_{i1} \\ \chi_{i1} \\ S_{i2} \\ \chi_{i2} \end{pmatrix}, \quad \psi_i^\dagger = \begin{pmatrix} \psi_R^\dagger & \psi_A^\dagger \end{pmatrix} = \begin{pmatrix} S_{i1}^* & \chi_{i1}^* & S_{i2}^* & \chi_{i2}^* \end{pmatrix} \tag{12}$$

are 4-dimensional super-vectors with ordinary (complex, commuting) $(S_{i1}, S_{i2})$ and Grassmanian (anti-commuting) $(\chi_{i1}, \chi_{i2})$ components,

$$\hat{K} = \begin{pmatrix} 1 & \\ & -1 \end{pmatrix}_{BF} = \mathrm{diag} \begin{pmatrix} 1 & -1 & 1 & -1 \end{pmatrix}, \tag{13}$$

$$\hat{L} = \begin{pmatrix} 1 & \\ & -1 \end{pmatrix}_{RA} = \mathrm{diag} \begin{pmatrix} 1 & 1 & -1 & -1 \end{pmatrix}, \tag{14}$$

$$\hat{J} = \begin{pmatrix} J_R & \\ & J_A \end{pmatrix}_{RA} = \mathrm{diag} \begin{pmatrix} J_R & J_R & J_A & J_A \end{pmatrix} \tag{15}$$

and $[d\psi] = \left[ d\psi_R d\psi_R^\dagger \right] \left[ d\psi_A d\psi_A^\dagger \right]$.

# 3 Functional integral and saddle point equations

Starting from Eq.(11) one needs to perform quite a number of mathematical calculations, which are described in details in the Section B of the Supplement. To put it briefly, the first step is to average over realizations of L'evy distributed matrix elements. Next step in a usual supersymmetric approach is to use Hubbard-Stratonovich transformation, which however is not useful in our case of the power-law tailed distributions. Instead, we use functional analog of the Hubbard-Stratonovich transform, which was proposed and described in details in [[2], see also [20]]. Following this approach (see also Sec. B2), the partition function (11) can be transformed into the following functional integral over functions $g(\psi, \psi^+)$ dependent on super-vectors $\psi$ and $\psi^+$:

$$Z \left( E, \omega, \hat{J} \right) = \int Dg \exp \left( S \left[ g \left( \psi, \psi^\dagger \right) \right] \right), \tag{16}$$

where the functional action $S \left[ g \left( \psi, \psi^\dagger \right) \right]$ is given by

$$S \left[ g \left( \psi, \psi^\dagger \right) \right] = N \ln \left\langle \int [d\psi] \exp \left( i\psi^\dagger \left( E\hat{L} + \frac{\Omega}{2} - \hat{J}\hat{K}\hat{L} - \zeta\hat{L} \right) \psi - g \left( \psi, \psi^\dagger \right) \right) \right\rangle_\zeta + \tag{17}$$

$$\frac{N}{2} \int [d\psi] [d\psi'] \, g \left( \psi, \psi^\dagger \right) \mathcal{I}^{-1} \left( \psi'^\dagger \hat{L} \psi \right) g \left( \psi', \psi'^\dagger \right).$$

where $\mathcal{I}(x) \equiv \frac{\sigma^\mu N^{1-\gamma}}{\Gamma\left(\frac{\mu}{2}+1\right)} \left[ x^\dagger x \right]^{\mu/2}$ and $\zeta$ corresponds to diagonal elements. Variable $\zeta$ stays for elements of diagonal matrix $H_D$ and its distribution is smooth at the scale of bandwidth $W$.

Due to the large prefactor $N$ in the action, one can perform the functional integration over $g(\psi, \psi^+)$ by the steepest descent method which leads to the self-consistency equation, those explicit form depends on the energy argument $\omega$:

$$g_\omega\left(\psi'^\dagger\psi', \psi'^\dagger\hat{L}\psi'\right) = \left\langle\left(\int [d\psi]\,\mathcal{I}\left(\psi'^\dagger\hat{L}\psi\right)\exp\left(i\psi^\dagger\left(E\hat{L} - \zeta\hat{L} + \frac{\Omega}{2}\right)\psi - g_\omega\left(\psi^\dagger\psi, \psi^\dagger\hat{L}\psi\right)\right)\right)\right\rangle_\zeta \tag{18}$$

As follows from the form of Eq.(18), its solution depends on two invariant objects: $\psi'^\dagger\psi'$ and $\psi'^\dagger\hat{L}\psi'$. Details of the solution of Eq.(18) are provided in Sec.C of the Supplement.

The key physical observation which helps to solve Eq.(18) goes as follows: $e^{-g_\omega(\psi^\dagger\psi, \psi^\dagger\hat{L}\psi)}$ is the characteristic function of a complex self-energy function $\Sigma$ of the operator $(\hat{H} - E)^{-1}$. Reduced functions of only single arguments, $e^{-g_\omega(0, \psi^\dagger\hat{L}\psi)}$ and $e^{-g_\omega(\psi^\dagger\psi, 0)}$, represent characteristic functions of real and imaginary part of the same self-energy, respectively. Now, the key point is that the full function $g_\omega\left(\psi^\dagger\psi, \psi^\dagger\hat{L}\psi\right)$ can be represented as a simple sum of two independent functions:

$$g_\omega\left(\psi^\dagger\psi, \psi^\dagger\hat{L}\psi\right) \approx g_\omega\left(0, \psi^\dagger\hat{L}\psi\right) + g_\omega\left(\psi^\dagger\psi, 0\right) \tag{19}$$

It means that real and imaginary parts of the self-energy $\Sigma$ are independently distributed. Physical reason for such an independence is that $\mathrm{Re}\,\Sigma(E)$ acquires relevant contributions from a broad range of energies $E \sim W$, while $\mathrm{Im}\,\Sigma(E)$ is determined by the close vicinity of $E$ only. This phenomenon was studied in details in Ref. [3]. The distribution of $\mathrm{Re}\,\Sigma$ was evaluated in our previous paper [20] where the average density of states was calculated. It leads to a slight renormalization of spectrum $\sim \frac{\sigma}{W}$ and can be omitted in the present problem. The reason can be seen in Eq.(18): integration over $d\zeta$ over the broad range $\sim W$ makes very small relevant values of $\psi^\dagger\hat{L}\psi \leq \frac{1}{W}$. In result, it is sufficient to work with $g_\omega\left(\psi^\dagger\psi, 0\right)$.

At sufficiently large $\omega$ saddle-point solution of the type of (19) is sufficient for the purpose of our calculations (precise criterion on the range of $\omega$ will be present below). The corresponding solution is described in Sec. C of the Supplement, the result reads a follows:

$$g_{\text{s.p.}}\left(\psi, \psi^\dagger\right)\Big|_{\psi^\dagger\hat{L}\psi=0} = g_\omega\left(\psi^\dagger\psi, 0\right) = \left[\Gamma_\omega\psi^\dagger\psi\right]^{\mu/2} \tag{20}$$

where function $\Gamma_\omega$ is determined by the transcendental equation

$$\Gamma_\omega^{\mu-1} = \Gamma_0^{\mu-1}\frac{\Gamma\left(\frac{\mu}{2}\right)}{\Gamma\left(2 - \frac{2}{\mu}\right)}\int_0^\infty dr L_{\mu/2}(r)\left[r - i\frac{\omega}{\Gamma_\omega}\right]^{1-\frac{\mu}{2}}, \tag{21}$$

and its zero-frequency limit $\Gamma_0$ is expressed via energy parameters $\sigma$ and $\Delta$ as follows:

$$\left[\frac{\Gamma_0}{2}\right]^{\mu-1} = \frac{\sigma^\mu}{\Delta N^\gamma}\frac{\sqrt{\pi}\Gamma\left(\frac{\mu-1}{2}\right)\Gamma\left(2 - \frac{2}{\mu}\right)}{\Gamma^2\left(\frac{\mu}{2}\right)} \tag{22}$$

with $\Gamma(x)$ in the R.H.S. being Euler Gamma-functions.

To meet the requirements of intermediate non-ergodic state one needs to apply the constraint $\Delta \ll \Gamma_0 \ll W$ in $N \to \infty$ limit (otherwise saddle point approximation is not valid). This will lead to inequalities: $N^{\frac{\gamma}{\mu}-1} < \frac{\sigma}{W} < N^{\frac{\gamma-1}{\mu}}$. However, numerical prefactor in (22)

strongly diverges at $\mu \to 1$ so one should be careful with the choice of specific parameters while doing numerical study.

Few remarks are in order now. First, we note that the form of the saddle-point solution (20) demonstrates a heavy-tail nature of distributions of $\operatorname{Im}\Sigma$ and $\operatorname{Im}G$. Second, we emphasize the appearance of a new energy scale $\Gamma_0$ determined by Eq.(22), see also Ref. [1]. In the Gaussian case $\mu = 2$ it gives just the value of the mini-band width [19], while for generic $1 < \mu < 2$ mini-band structure is more complicated, it is characterized by a distribution of widths which is characterized by the parameter given by Eq.(22); the same equation for $\Gamma_0$ was obtained in Ref. [3]. Third, at nonzero $\omega$ the function $\Gamma_\omega$ is complex, with $\operatorname{Im}\Gamma_\omega < 0$; this feature is related to the analytic properties of the DoS correlation function and it will be important later in Sec. 4.

At high frequencies $\omega \geq \Gamma_0$ the function $\Gamma_\omega$ can be obtained from Eq.(21) and behaves as

$$\frac{\Gamma_{\omega \to \infty}}{\Gamma_0} \sim \left|\frac{\omega}{\Gamma_0}\right|^{\frac{2}{\mu}-1} \left(\cos\left[\frac{\pi(2-\mu)}{2\mu}\right] - i\sin\left[\frac{\pi(2-\mu)}{2\mu}\right]\right) \left[\frac{\Gamma\left(\frac{\mu}{2}\right)}{\Gamma\left(2-\frac{2}{\mu}\right)}\right]^{\frac{2}{\mu}}. \tag{23}$$

Since L'evy distribution degenerates into a delta function at $\mu = 2$, $\Gamma_\omega$ becomes real constant $\Gamma_\omega = \Gamma_0$ regardless of $\omega$. On the other hand, at $\omega = 0$ saddle-point solution (20) is not unique: it belongs to the whole manifold of solutions those actions coincide. In result, to obtain physical quantities at low $\omega$ one should integrate over the whole saddle-point manifold, as it was done in Ref. [19] for Gaussian RP model. General solution that belongs to the saddle-point manifold can be written in the form

$$g_0\left(\psi^\dagger \hat{T}^\dagger \hat{T}\psi, \psi^\dagger \hat{L}\psi\right) \equiv g_T\left(\psi^\dagger \psi, \psi^\dagger \hat{L}\psi\right) \tag{24}$$

where $\hat{T}$ is the 4-dimensional super-matrix that rotates super-vectors $\psi$ and $\psi^+$. It obeys the symmetry property $\hat{T}^\dagger \hat{L}\hat{T} \equiv \hat{L}$.

In the next Section we will show that unique high-$\omega$ solution (20) is applicable at $\omega \gg E_{Th} \sim \sqrt{\Delta\Gamma_0}$ while integration over saddle-point manifold (24) can be employed at $\omega \ll \Gamma_0$. Since we always have $\Delta \ll \Gamma_0$, the combination of both approaches cover the whole range of frequencies we are interested in.

# 4 Correlation function results and asymptotics

In this section we calculate the DoS correlation function and discuss its properties. The main expression for the correlation function follows from (10) and (16):

$$R(E,\omega) = \frac{1}{2} + \frac{\Delta^2}{8\pi^2}\operatorname{Re}\int D[g]\left[\frac{\partial^2 S[g]}{\partial J_A \partial J_R} + \frac{\partial S[g]}{\partial J_R}\frac{\partial S[g]}{\partial J_A}\right]e^{S[g]}\Bigg|_{J_R, J_A = 0}. \tag{25}$$

where the action $S[g]$ is defined in Eq.(17). In the saddle-point approximation $g\left(\psi^\dagger, \psi\right)$ should be substituted by the solution ((20)). Quadratic over $g_\omega(\psi, \psi^+)$ term in the action does not depend on the sources $J_{A,R}$, it is also invariant under $\psi \to \hat{T}\psi$ transformations. Supersymmetry of this term means that it does not contribute to the action on the saddle-point manifold (see integration theorems in Refs. [24],[25] or Supplementary material [C]).

The only important term left in action is

$$S\left[g_{\text{s.p.}}\right] = N \ln \left\langle \int [d\psi] \exp \left( i\psi^\dagger \left( E\hat{L} + \frac{\Omega}{2} - \hat{J}\hat{K}\hat{L} - \zeta\hat{L} \right) \psi - g_{\text{s.p.}} \left( \psi, \psi^\dagger \right) \right) \right\rangle_\zeta \quad (26)$$

where $g_{\text{s.p.}}$ is saddle-point solution of (18). Supersymmetrical part of this expression should be equal to unity and the other part is assumed to be small, so that one can expand the logarithm to get

$$S\left[g_{\text{s.p.}}\right] = N \left\langle \left\{ \int [d\psi] \exp \left( i\psi^\dagger \left( E\hat{L} + \frac{\Omega}{2} - \hat{J}\hat{K}\hat{L} - \zeta\hat{L} \right) \psi - g_{\text{s.p.}} \left( \psi, \psi^\dagger \right) \right) \right\}_{\hat{J},\hat{T}\neq 0} \right\rangle_\zeta \quad (27)$$

Integration over $d\zeta$ in Eq.(27 goes smoothly over broad range of energies $\sim W$ which leads effectively to the restriction of $\psi^\dagger\hat{L}\psi$ being very small (by the same logics as was described in the analysis of the saddle-point solution above). In result, one can employ $g_{\text{s.p.}} \left( \psi, \psi^\dagger \right) \Big|_{\psi^\dagger\hat{L}\psi=0}$ from the solution (20) to get

$$S\left[g_{\text{s.p.}}\right] = N \left\langle \left\{ \int [d\psi] \exp \left( i\psi^\dagger \left( E\hat{L} + \frac{\Omega}{2} - \hat{J}\hat{K}\hat{L} - \zeta\hat{L} \right) \psi - g_{\text{s.p.}} \left( \psi, \psi^\dagger \right) \Big|_{\psi^\dagger\hat{L}\psi=0} \right) \right\}_{\hat{J},\hat{T}\neq 0} \right\rangle_\zeta$$
$$(28)$$

Further analysis differs for small and large $\omega$. First we consider high-frequency region within saddle-point approximation; the domain of applicability of these results becomes clear by comparison with results of exact calculation provided later for the low frequency region.

## 4.1 High frequencies $\omega \gg E_{th} \equiv \sqrt{\Delta\Gamma/2\pi}$

In the high-$\omega$ limit (parameter $\Gamma$ is defined in Sec.4.2) one employs $g_\omega \left( \psi^\dagger\psi, 0 \right)$ solution. It is sufficient to calculate saddle-point action as function of the sources:

$$S\left[g_\omega\right] = N \left\langle \left\{ \int [d\psi] \exp \left( i\psi^\dagger \left( E\hat{L} + \frac{\Omega}{2} - \hat{J}\hat{K}\hat{L} - \zeta\hat{L} \right) \psi - \left[ \Gamma_\omega\psi^\dagger\psi \right]^{\mu/2} \right) \right\}_{\hat{J}\neq 0} \right\rangle_\zeta \quad (29)$$

Recalling properties of one-sided L'evy distribution and definition of super-determinant, we find

$$S\left[g_\omega\right] = N \int_0^\infty dr L_{\frac{\mu}{2}}(r) \left\langle \left\{ \text{Sdet}^{-1} \left( E - \zeta + \left( \frac{\Omega}{2} + i\Gamma_\omega r \right) \hat{L} - \hat{J}\hat{K} \right) \right\}_{\hat{J}\neq 0} \right\rangle_\zeta \quad (30)$$

At this stage it is useful to define the corresponding Green function

$$\hat{G} \equiv \left( E - \zeta + \left( \frac{\Omega}{2} + i\Gamma_\omega r \right) \hat{L} \right)^{-1} = \frac{E - \zeta - \left( \frac{\Omega}{2} + i\Gamma_\omega r \right) \hat{L}}{(E-\zeta)^2 - \left( \frac{\Omega}{2} + i\Gamma_\omega r \right)^2} \quad (31)$$

Employing exact relation for super-determinants, $\ln \text{Sdet}\hat{A} = \text{Str}\ln\hat{A}$ one can expand action in Eq.(30) it over sources $J_{R,A}$:

$$S\left[g_\omega\right] = N \int dr L_{\frac{\mu}{2}}(r) \left[ \left\langle \text{Str}\left[\hat{G}\hat{J}\hat{K}\right] \right\rangle_\zeta + \frac{1}{2} \left\langle \text{Str}^2\left[\hat{G}\hat{J}\hat{K}\right] \right\rangle_\zeta + \frac{1}{2} \left\langle \text{Str}\left[\hat{G}\hat{J}\hat{K}\hat{G}\hat{J}\hat{K}\right] \right\rangle_\zeta \right] \quad (32)$$

Distribution $P_D(\zeta)$ is a very slow function of $\zeta$, as compared to $\zeta$-dependence of the Green function $\hat{G}$ defined in (31), so it is possible to use approximation $P_D(\zeta) \approx P_D(E) \sim W^{-1}$. Performing integration near the pole (with the use of the fact that $\operatorname{Im}\Gamma_\omega < 0$) and also the relation $P_D(E)N = \Delta^{-1}$, we arrive to

$$S[g_\omega] = -i\frac{\pi}{\Delta}\left\{2(J_R - J_A) - 8J_R J_A \int dr \frac{L_{\frac{\mu}{2}}(r)}{\Omega + 2i\Gamma_\omega r}\right\}, \tag{33}$$

Substitution of (33) into (25) gives final result in the form

$$R(\omega) = 1 + \frac{\Delta}{\pi}\int_0^\infty dr \frac{L_{\frac{\mu}{2}}(r) \cdot 2r\operatorname{Re}\Gamma_\omega}{[\omega - 2r\operatorname{Im}\Gamma_\omega]^2 + [2r\operatorname{Re}\Gamma_\omega]^2} \tag{34}$$

In the limit $\mu \to 2$ the above result coincide with the one obtained in Ref. [19] for Gaussian RP model at high $\omega$. For general $\mu$ similar result was obtained in Ref. [3] where local DoS correlation function $C(\omega)$ was obtained by means of cavity equation; the relation between these results is as follows: $R(\omega) - 1 = 2^{\mu/2}\Delta \cdot C(\omega)$. The difference in numerical coefficients is due to slightly different models: while we consider Hermitian matrix ensemble with complex off-diagonal elements, the function $C(\omega)$ is calculated in [3] for real matrix ensemble. At high frequencies the main asymptotics is given by the power-law

$$R(\omega) = 1 + \frac{\Delta}{\pi\Gamma_0}\frac{2^{\mu/2}\Gamma\left(\frac{\mu}{2}\right)\Gamma\left(\frac{\mu}{2}+1\right)}{\Gamma\left(2-\frac{2}{\mu}\right)}\left(\frac{\Gamma_0}{\omega}\right)^\mu \tag{35}$$

We present details of this calculation in Appendix D.

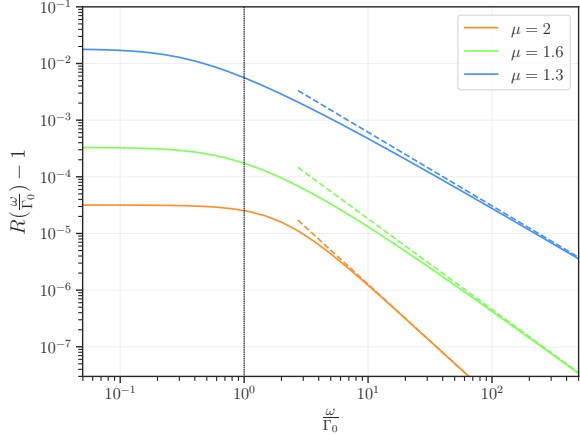

Figure 2: Correlation function $R(\omega) - 1$ in log-log scale (see Eq.(34)). We choose $\gamma = 1.1$, $\sigma/W = 0.02$, $N = 10^7$. Dashed lines correspond to the asymptotic solution provided in Eq.(35)

.

## 4.2   Domain $\omega \ll \Gamma_0$

Now we can use expansion over parameter $\frac{\omega}{\Gamma_0} \ll 1$. We will keep nonzero $\omega$ in the action (28) only and replace $g_{\text{s.p.}}\left.\left(\psi, \psi^\dagger\right)\right|_{\psi^\dagger \hat{L}\psi = 0}$ used in the previous Sec.4.1 by saddle-point manifold

$g_T \left( \psi^\dagger \psi, 0 \right)$ parametrized by the rotation matrix $\hat{T}$. One should remember the definition $\omega + i0 \equiv \Omega$, so that if $\omega = 0$ than $\Omega = i0$ to maintain the convergence of integrals.

After inverse field transformation $\psi \to \hat{T}^{-1}\psi$ the action acquires the form

$$S\left[g_T\right] = \frac{1}{\Delta} \int dr L_{\frac{\mu}{2}}\left(r\right) \left\langle \left\{ \mathrm{Sdet}^{-1} \left( E + \left( \frac{\Omega}{2}\hat{T}\hat{T}^\dagger + i\Gamma_0 r \right) \hat{L} - \hat{T}\hat{J}\hat{K}\hat{L}\hat{T}^\dagger\hat{L} - \zeta \right) \right\}_{\hat{J},\hat{T}\neq 0} \right\rangle_\zeta \tag{36}$$

and integration over functions $D\left[g\right]$ in (25) is replaced by the integration over $\hat{T}$ matrices. Matrix $\hat{T}$ is closely connected with Efetov matrix $\hat{Q}$ as $\hat{T}^\dagger\hat{T} = \hat{L}\hat{Q}$ (see Supplement E). Further procedure is similar to the one used in the previous subsection. First of all one performs an expansion over $\frac{\Omega}{\Gamma_0}$ and $\hat{J}$. Then, using the same tricks as in (32)-(33) one obtains an intermediate result in terms of $\hat{Q}$ matrices (remember that $\mathrm{Str}\left[a\hat{1} + b\hat{L}\right] = 0$ for any numbers $a, b$).

$$S_0\left(\hat{T}, \hat{J}\right) = \frac{i\pi}{\Delta}\mathrm{Str}\left( \frac{\Omega}{2}\hat{L}\hat{Q} - \hat{J}\hat{K}\hat{Q} \right) + \tag{37}$$

$$\frac{\pi}{2\Delta\Gamma} \left\{ \mathrm{Str}\left( \hat{J}^2 - \Omega\hat{J}\hat{K} \right) - \mathrm{Str}\left( \hat{J}\hat{K}\hat{Q}\hat{J}\hat{K}\hat{Q} - \Omega\hat{J}\hat{K}\hat{Q}\hat{L}\hat{Q} + \frac{\Omega^2}{4}\hat{L}\hat{Q}\hat{L}\hat{Q} \right) + \right.$$

$$\left. \mathrm{Str}^2\left[ \hat{J}\hat{K} \right] - \mathrm{Str}^2\left( \hat{J}\hat{K}\hat{Q} - \frac{\Omega}{2}\hat{L}\hat{Q} \right) \right\}$$

Finally, the key parameter $\Gamma$ is determined as follows:

$$\Gamma \equiv \left[ \int dr \frac{L_{\mu/2}\left(r\right)}{2r\Gamma_0} \right]^{-1} = \frac{2\Gamma_0}{\Gamma\left( \frac{2}{\mu} + 1 \right)}. \tag{38}$$

Using the relation $\hat{J} = J_R \frac{\hat{L}+1}{2} + J_A \frac{1-\hat{L}}{2}$, we calculate the derivatives and obtain the following terms in the action (25) :

$$\left. S\left[g_T\right] \right|_{J_{R,A}=0} = \frac{i\pi\Omega}{2\Delta}\mathrm{Str}\left( \hat{L}\hat{Q} \right) - \frac{\pi\Omega^2}{8\Delta\Gamma}\left( \mathrm{Str}\left( \hat{L}\hat{Q}\hat{L}\hat{Q} \right) + \mathrm{Str}^2\left( \hat{L}\hat{Q} \right) \right) \tag{39}$$

,

$$\frac{\partial^2 S\left[g_T\right]}{\partial J_A \partial J_R} = \frac{\pi}{\Delta\Gamma}\left[ \mathrm{Str}\left( \hat{U}_-\hat{U}_+ \right) + \mathrm{Str}\left( \hat{U}_- \right)\mathrm{Srt}\left( \hat{U}_+ \right) + 4 \right] \tag{40}$$

,

$$\frac{\partial S\left[g_T\right]}{\partial J_R}\frac{\partial S\left[g_T\right]}{\partial J_A} = -\left[ \frac{i\pi}{\Delta}\mathrm{Str}\left( \left[\hat{U}_+\right] \right) + \frac{\pi\Omega}{2\Delta\Gamma}\left( 2 - \mathrm{Str}\left( \hat{U}_+\hat{L}\hat{Q} \right) - \mathrm{Str}\left( \hat{U}_+ \right)\mathrm{Str}\left( \hat{L}\hat{Q} \right) \right) \right] \times$$

$$\left[ \frac{i\pi}{\Delta}\mathrm{Str}\left( \hat{U}_- \right) + \frac{\pi\Omega}{2\Delta\Gamma}\left( 2 - \mathrm{Str}\left( \hat{U}_-\hat{L}\hat{Q} \right) - \mathrm{Str}\left( \hat{U}_- \right)\mathrm{Str}\left( \hat{L}\hat{Q} \right) \right) \right], \tag{41}$$

where $\hat{U}_+ \equiv \frac{\hat{L}+1}{2}\hat{K}\hat{Q}$ and $\hat{U}_- = \frac{\hat{L}-1}{2}\hat{K}\hat{Q}$.

The relation (38) above means that the quantity which should be averaged over Levy distribution is the *inverse* mini-band width $1/r$, which is equivalent to the *decay time* from

the mini-band. Evaluation of integrals like the one present in Eq.(38) is discussed in details in Ref. [3]. The quantity $\Gamma$ is similar to the one defined in [19] for the Gaussian RP model and coincides with it at $\mu = 2$.

Now we should integrate all manifold of $\hat{Q}$ in (40)-(39). Unitary matrix $\hat{Q}$ is parameterized in a standard way using Efetov parametrization (see Supplement E). Two different energy scales appear in (39). First term contains mean level spacing $\Delta$ and leads to the oscillations at $\omega \sim \Delta$, while the second one defines Thouless energy $E_{th} \equiv \sqrt{\frac{\Delta\Gamma}{2\pi}}$, as an energy where typical GUE oscillations become exponentially suppressed. Combining all terms, we find the final integral expression for the correlation function at $\omega \ll \Gamma$:

$$R\left(E,\omega\right) = 1 + \frac{\Delta}{\pi\Gamma} +$$
$$\text{Re} \int_1^\infty d\lambda_B \int_{-1}^1 d\lambda_F \left[\left(1 + \frac{2i\Omega}{\Gamma}\lambda_B\right)^2 + \frac{\Delta}{\pi\Gamma}\frac{\lambda_B}{\lambda_B - \lambda_F}\right] \exp\left(\frac{i\pi\Omega}{\Delta}\left(\lambda_B - \lambda_F\right) - \frac{\pi\Omega^2}{\Delta\Gamma}\lambda_B\left(\lambda_B - \lambda_F\right)\right)$$
$$(42)$$

Double integral in Eq.(42) can be further simplified using large parameter $\omega/\Delta \gg 1$ and we find (see Supplement, Sec.F for details):

$$R\left(E,\omega\right) \approx 1 + \frac{\Delta}{\pi\Gamma} - \frac{\Delta^2}{2\pi^2\omega^2}\left(1 - \cos\left(\frac{2\pi\omega}{\Delta}\right)\exp\left(-\frac{2\pi\omega^2}{\Delta\Gamma}\right)\right). \tag{43}$$

The above result coincides with the one obtained for the Gaussian-RP model up to renormalization of the mini-band width $\Gamma$. At low frequencies $\omega \ll E_{th}$ we get from Eq.(43) a simplified expression

$$R\left(E,\omega\right) = 1 - \frac{\sin^2\left(\frac{\pi\omega}{\Delta}\right)}{\left(\frac{\pi\omega}{\Delta}\right)^2} + \frac{2\Delta}{\pi\Gamma}\sin^2\left(\frac{\pi\omega}{\Delta}\right), \tag{44}$$

which coincides with GUE limit when $\Gamma/\Delta \to \infty$. The whole behavior of $R(\omega)$ at all frequencies is shown in Fig.(3).

## 5   Discussion and Conclusions

We calculated energy level correlation function $R(\omega)$ for Lévy Rosenzveig-Porter ensemble by means of supersymmetry method. Our major new result provided by Eqs.(43, 38) refers to low-frequency range $\omega \le E_{Th}$. Functional form of Eq.(43) reproduces the one known for Gaussian RP model [19], while inverse of effective mini-band width $1/\Gamma$ is given by the average over Lévy distribution of local decay times, as follows from Eq.(38). In the high-frequency domain our result is given by Eqs.(34,21) and is in agreement with the result of Ref. [3] for the correlation function of local density of states $C(\omega)$.

The major qualitative difference between Gaussian RP and Lévy-RP ensembles is that the first one can be described in terms of average Green functions $G(E)$ and self-energies $\Sigma(E)$, while in the Lévy-RP case one is forced to consider non-trivial probability distributions for both Green function and self-energy. Moreover, the mini-band width $\Gamma_0$ known for Gaussian RP ensemble becomes a random quantity in the Lévy-PR model, as can be observed with

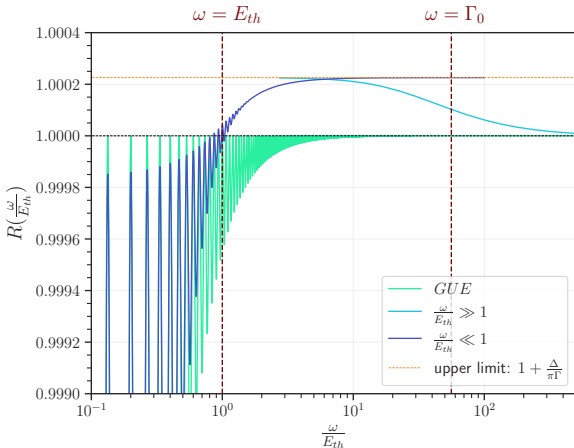

Figure 3: Correlation function obtained from (34) and (43) asymptotics. $\mu = 1.5$, $\quad \sqrt{\Delta/\Gamma_0} \approx 0.035$, $\quad \sigma \approx 0.023W$, $\quad \gamma = 1.1$, $\quad N = 10^5$ (see Eq. (22)). Solutions (34) and (43) have an upper limit equal to $1 + \frac{\Delta}{\pi\Gamma}$. Contrary to the case of GUE correlation function which never exceeds 1 (levels only repel each other), Levy-RP model demonstrate weak long-range level attraction at $\omega > E_{th}$.

Eq.(38): effective width of a mini-band $\Gamma$ is found to be an inverse of a realization-dependent decay time $1/r\Gamma_0$ over Lévy distribution.

Long power-law tail in the distribution of off-diagonal matrix elements makes mini-bands of Lévy-RP ensemble different from their Gaussian-RP counterparts which are compact in the values of bare energies (diagonal matrix elements $\zeta_i$). Since there is a quite considerable probability to find abnormally large matrix element $H_{nm}$ in the Lévy-RP case, here mini-bands are partially overlapping in the energy space.

On a technical side, our results demonstrate that field-theoretic approach based on super-symmetry can be efficiently employed for the analysis of systems described by random Hamiltonian with a heavy-tailed distributions. We expect that such an approach might be useful for the analysis of spatially extended systems with internal structure, like the one studied in Ref. [26] but with a Lévy distribution of hopping matrix elements.

# 6    Acknowledgments

The authors are grateful to Yan Fyodorov, Vladimir Kravtsov, Pavel Ostrovsky and Marco Tarzia for many useful discussions.

# A  Green functions and supersymmetric field theory

## A.1  Supersymmetric field theory

Representation of Green functions in supersymmetric field theory is based on the property of determinants that

$$\ln \det \hat{A} = \mathrm{Tr} \ln \hat{A} \Rightarrow \mathrm{Tr}\left[\hat{A}\right]^{-1} = \frac{1}{2}\frac{\partial}{\partial J}\frac{\det\left(\hat{A}+J\right)}{\det\left(\hat{A}-J\right)}\bigg|_{J=0}. \tag{45}$$

Whereas $\langle \mathrm{Tr} G_R \mathrm{Tr} G_A \rangle$-type function represents as following

$$\left\langle \mathrm{Tr}\left[E + \frac{\Omega}{2} - \hat{H}\right]^{-1} \mathrm{Tr}\left[E - \frac{\Omega}{2} - \hat{H}\right]^{-1} \right\rangle_{\hat{H}} =$$

$$= \frac{\partial^2}{\partial J_R \partial J_A}\left\langle \frac{\det\left(E - \hat{H} + \frac{\Omega}{2} + J_R\right)\det\left(E - \hat{H} - \frac{\Omega}{2} + J_A\right)}{\det\left(E - \hat{H} + \frac{\Omega}{2} - J_R\right)\det\left(E - \hat{H} + \frac{\Omega}{2} - J_A\right)} \right\rangle_{\hat{H}} \tag{46}$$

Using the basic properties of Gaussian integrals(for commutative and anti-commutative variables):

$$\int e^{-\vec{\chi}^\dagger \hat{A} \vec{\chi}} d\vec{\chi}^\dagger d\vec{\chi} = \det\left(\frac{\hat{A}}{2\pi}\right), \quad \int e^{-\vec{S}^\dagger \hat{A} \vec{S}} d\vec{S}^\dagger d\vec{S} = \frac{1}{\det\left(\frac{\hat{A}}{2\pi}\right)}$$

one arrives to the result (10).

**Remark:** The sign of anti-commutative variables is not matter for convergence of the integral, however it is necessary choose right the sign of commuting variables.

# B  Derivation of (16),(17)

## B.1  Averaging over off-diagonal matrix elements

We start by the averaging of the partition function (11) over the random entries of $\hat{H}$

$$Z\left(E, \omega, \hat{J}\right) = \exp\left(i\sum_n^N \psi_n^\dagger \left[E\hat{L} + \frac{\Omega}{2} - \hat{J}\hat{K}\hat{L}\right]\psi_n\right) \times \tag{47}$$

$$\exp\left(\ln\left\langle \exp\left(-i\sum_{n,m}^N \psi_n^\dagger \left([H_L]_{nm} + \delta_{nm}[H_D]_{nn}\right)\hat{L}\psi_m\right)\right\rangle_{\hat{H}_L, \hat{H}_D}\right)$$

Gaussian element typical value $\sim W$(assumed much larger than L'evy diagonals typical value) so that it is reasonable to leave only $[H_D]_{nm}$ on diagonal. This splits the averaging $\langle ... \rangle$ into two independent parts. The Hermitian property of the matrix $\hat{H}_L$ allows separate the rest of sum into independent symmetrical entrees, resulting in

$$Z\left(E, \omega, \hat{J}\right) = \left\langle \exp\left(i \sum_n^N \psi_n^\dagger \left[E\hat{L} + \frac{\Omega}{2} - \hat{J}\hat{K}\hat{L} - [H_D]_{nn}\right] \psi_n\right)\right\rangle_{\hat{H}_D} \times \tag{48}$$

$$\ln\left\langle \exp\left(-i \sum_{m<n}^N \left[\psi_n^\dagger [H_L]_{nm} \hat{L}\psi_m + \psi_m^\dagger [H_L]_{nm}^* \hat{L}\psi_n\right]\right)\right\rangle_{\hat{H}_L}$$

Since the symmetrical pairs of the matrix elements are independent, the second line in the above equation can be rewritten as follows:

$$\ln\left\langle \exp\left(-i \sum_{m<n}^N \left[\psi_n^\dagger [H_L]_{nm} \hat{L}\psi_m + \psi_m^\dagger [H_L]_{nm}^* \hat{L}\psi_n\right]\right)\right\rangle_{\hat{H}_L} = \tag{49}$$

$$= \frac{1}{2} \sum_{n\neq m}^N \ln\left\langle \exp\left(-i \left[\psi_n^\dagger [H_L]_{nm} \hat{L}\psi_m + \psi_m^\dagger [H_L]_{nm}^* \hat{L}\psi_n\right]\right)\right\rangle_{\hat{H}_L}.$$

Furthermore, because there are $\sim N$ diagonal entries and $\sim N^2$ off-diagonal ones, one can replace $\sum_{m\neq m}$ by $\sum_{m,n}$. Later on, using the fact that off-diagonal matrix elements $[H_L]_{nm}$ are smaller that diagonal ones by the factor $N^\gamma$, one can use the following approximation

$$\sum_{n,m}^N \ln\left\langle \exp\left(-i \left[\psi_n^\dagger [H_L]_{nm} \hat{L}\psi_m + \psi_m^\dagger [H_L]_{nm}^* \hat{L}\psi_n\right]\right)\right\rangle_{\hat{H}_L} = \tag{50}$$

$$\sum_{n,m}^N \ln\left[1 + \left\langle \exp\left(-i \left[\psi_n^\dagger [H_L]_{nm} \hat{L}\psi_m + \psi_m^\dagger [H_L]_{nm}^* \hat{L}\psi_n\right]\right) - 1\right\rangle_{\hat{H}_L}\right] \approx$$

$$\approx \sum_{n,m}^N \left\langle \exp\left(-i \left[\psi_n^\dagger [H_L]_{nm} \hat{L}\psi_m + \psi_m^\dagger [H_L]_{nm}^* \hat{L}\psi_n\right]\right) - 1\right\rangle_{\hat{H}_L} \equiv \frac{1}{2N} \sum_{n,m}^N \mathcal{I}\left(\psi_n^\dagger \hat{L}\psi_m\right).$$

Let us now denote $[H_L]_{nm} \equiv he^{i\theta}$ and $\psi_n^\dagger \hat{L}\psi_m \equiv t$, so that $\psi_n^\dagger [H_L]_{nm} \hat{L}\psi_m + \psi_m^\dagger [H_L]_{nm}^* \hat{L}\psi_n \equiv h\left(te^{i\theta} + t^\dagger e^{-i\theta} - i0\right)$, where $-i0$ ensures convergence of the integral 50. The object $\mathcal{I}(t)$ entering last line of Eq.(50) can be rewritten as

$$\mathcal{I}(t) = 2N \int_{-\pi}^\pi \frac{d\theta}{2\pi} \int_0^\infty \frac{d\left[h^2\right]}{2} P_L\left(h^2\right) \left(e^{-ih\left[te^{i\theta} + t^\dagger e^{-i\theta} - i0\right]} - 1\right). \tag{51}$$

Using normalization conditions, 6 and following the calculations in A.1 Appendix of [3] paper one can proceed to the following form:

$$\mathcal{I}(t) = \frac{2\sigma^\mu \Gamma(-\mu)}{N^{\gamma-1}\Gamma\left(-\frac{\mu}{2}\right)} \int_{-\pi}^\pi \frac{d\theta}{2\pi} \left(i\left[e^{i\theta}t + e^{-i\theta}t^\dagger\right] + 0\right)^\mu, \tag{52}$$

where constant follows from normalization. To calculate the $\theta$ integral one can use its independence on the phase of $t, t^\dagger$:

$$\int_{-\pi}^\pi \frac{d\theta}{2\pi} \left(i\left[e^{i\theta}t + e^{-i\theta}t^\dagger\right] + 0\right)^\mu = |t|^\mu \int_{-\pi}^\pi \frac{d\theta}{2\pi} \left(0 + 2i\cos\theta\right)^\mu = |2t|^\mu \frac{\cos\left(\frac{\pi\mu}{2}\right) B\left(\frac{1}{2}, \frac{1+\mu}{2}\right)}{\pi} \tag{53}$$

Using the expression, one obtains the following result of the averaging over Lévy distribution:

$$\mathcal{I}(t) = -\frac{\sigma^\mu |t|^\mu}{N^{\gamma-1}\Gamma\left(1+\frac{\mu}{2}\right)} \tag{54}$$

In result, we find

$$\ln\left\langle \exp\left(-i\sum_{n,m}^N \psi_n^\dagger \left[H_L\right]_{nm} \hat{L}\psi_m\right)\right\rangle_{\hat{H}_L} \approx -\frac{1}{2N}\sum_{n,m}^N \frac{\sigma^\mu \left[\psi_n^\dagger \hat{L}\psi_m \psi_m^\dagger \hat{L}\psi_n\right]^{\mu/2}}{N^{\gamma-1}\Gamma\left(1+\frac{\mu}{2}\right)}. \tag{55}$$

$$Z\left(E,\omega,\hat{J}\right) = \tag{56}$$

$$\left\langle \int [d\psi]\exp\left(i\sum_n^N \psi_n^\dagger \hat{L}\left(E+\frac{\Omega}{2}\hat{L}-\hat{J}\hat{K}-[H_D]_{nn}\right)\psi_n - \frac{1}{2N}\sum_{n,m}^N \frac{\sigma^\mu \left[\psi_n^\dagger \hat{L}\psi_m \psi_m^\dagger \hat{L}\psi_n\right]^{\mu/2}}{N^{\gamma-1}\Gamma\left(1+\frac{\mu}{2}\right)}\right)\right\rangle_{\hat{H}_D}$$

## B.2 Functional Hubbard-Stratonovich transformation

An obvious difficulty that still remains is the non-analytic power $\mu$ of $\psi_n^\dagger \hat{L}\psi_m \psi_m^\dagger \hat{L}\psi_n$ in the functional (instead of the quadratic term arising for the Gaussian distribution). This non-analycity encodes the fat tails in the distribution which, in their turn, determine the peculiar physical properties of the system. In order to decouple the super-vectors we use *the functional Hubbard-Stratonovich(H-S) transformation* [2] instead of the usual one. Generalized expression looks as follows:

$$\exp\left(\frac{1}{2N}\int [d\psi]\left[d\psi'\right] v\left(\psi\right) C\left(\psi,\psi'\right) v\left(\psi'\right)\right) =$$

$$\int Dg \exp\left(-\frac{N}{2}\int [d\psi]\left[d\psi'\right] g\left(\psi\right) C^{-1}\left(\psi,\psi'\right) g\left(\psi'\right) + \int [d\psi] g\left(\psi\right) v\left(\psi\right)\right), \tag{57}$$

where $C\left(\psi,\psi'\right)$, $v(\psi)$ and $g(\psi)$ are some functions or fields.

The advantage of this method and formal derivation was discussed in details in our previous paper [20] dedicated to the calculation of the average DoS by the same method. Hence, only the final formulae will be provided in the present paper:

$$\exp\left(-\frac{1}{2N}\cdot\frac{\sigma^\mu N^{1-\gamma}}{\Gamma\left(\frac{\mu}{2}+1\right)}\sum_{n,m}^N \left[\psi_n^\dagger \hat{L}\psi_m \psi_m^\dagger \hat{L}\psi_n\right]^{\mu/2}\right) = \tag{58}$$

$$\int \mathcal{D}g\exp\left(\frac{N}{2}\int [d\psi]\left[d\psi'\right] g\left(\psi,\psi^\dagger\right)\left\{\frac{\sigma^\mu N^{1-\gamma}}{\Gamma\left(\frac{\mu}{2}+1\right)}\left[\psi^\dagger \hat{L}\psi' \psi'^\dagger \hat{L}\psi\right]^{\mu/2}\right\}^{-1} g\left(\psi',\psi'^\dagger\right) - Ng\left(\psi,\psi^\dagger\right)\right)$$

Here we introduced functional integral over functions of super-fields $g\left(\psi,\psi^\dagger\right)$. Combining 58 with the previous expression (56) leads to the equations (16,17) for the partition function. Factor $N$ in the exponent comes due to $N$ independent integrations over $\psi_n, \psi_n^+$.

# C  Saddle-point equation and its solution

## C.1  Derivation of the saddle-point equation

Equating to zero variation of the action ((17)) over $\delta g\left(\psi, \psi^\dagger\right)$, one obtains the following integral equation for the saddle-point:

$$g_{\text{s.p.}}\left(\psi'^\dagger, \psi'\right) = \frac{\left\langle \int [d\psi]\, \mathcal{I}\left(\psi'^\dagger \hat{L}\psi\right) \exp\left(i\psi^\dagger\left(E\hat{L} - \zeta\hat{L} + \frac{\Omega}{2}\right)\psi - g_{\text{s.p.}}\left(\psi^\dagger, \psi\right)\right)\right\rangle_\zeta}{\left\langle \int [d\psi] \exp\left(i\psi^\dagger\left(E\hat{L} - \zeta\hat{L} + \frac{\Omega}{2}\right)\psi - g_{\text{s.p.}}\left(\psi^\dagger, \psi\right)\right)\right\rangle_\zeta} \qquad (59)$$

where $\mathcal{I}(x) \equiv \frac{\sigma^\mu N^{1-\gamma}}{\Gamma\left(\frac{\mu}{2}+1\right)}\left[x^\dagger x\right]^{\mu/2}$. The structure of Eq.(59) indicates that its solution is a function of two invariants: $g_{\text{s.p.}}\left(\psi^\dagger, \psi\right) = g_\omega\left(\psi^\dagger\psi, \psi^\dagger\hat{L}\psi\right)$. Once we search for the solution in this form, the integrand of the integral in the denominator is found to be invariant under the super-unitary transformations $\psi_{R,A} \to \hat{U}\psi_{R,A}$, $\psi = \left(\begin{array}{cc} \psi_R & \psi_A \end{array}\right)^T$, thus it is equal to unity. Therefore the final form of the saddle-point equation is

$$g_\omega\left(\psi'^\dagger\psi', \psi'^\dagger\hat{L}\psi'\right) = \left\langle \int [d\psi]\, \mathcal{I}\left(\psi'^\dagger\hat{L}\psi\right)\exp\left(i\psi^\dagger\left(E\hat{L} - \zeta\hat{L} + \frac{\Omega}{2}\right)\psi - g_\omega\left(\psi^\dagger\psi, \psi^\dagger\hat{L}\psi\right)\right)\right\rangle_\zeta.$$
$$(60)$$

At $\Omega = 0$ the saddle-point solution becomes

$$g_\omega\left(\psi^\dagger\psi, \psi^\dagger\hat{L}\psi\right)\bigg|_{\omega=0} \equiv g_0\left(\psi^\dagger\psi, \psi^\dagger\hat{L}\psi\right). \qquad (61)$$

Actually at $\Omega = 0$ the whole saddle manifold of solutions exist, which can be parametrized by the rotation matrix $\hat{T}$ subject to the condition $\hat{T}^\dagger\hat{L}\hat{T} = \hat{L}$:

$$\psi \to \hat{T}\psi, \quad g_T\left(\psi, \psi^\dagger\right) \equiv g_0\left(\psi^\dagger\hat{T}^\dagger\hat{T}\psi, \psi^\dagger\hat{L}\psi\right). \qquad (62)$$

Saddle-manifold solutions of this kind obey the equation

$$g_T\left(\psi', \psi'^\dagger\right) = \left\langle \int [d\psi]\, \mathcal{I}\left(\psi'^\dagger\hat{L}\psi\right)\exp\left(i\psi^\dagger\left(E - \zeta\right)\hat{L}\psi - g_T\left(\psi, \psi^\dagger\right)\right)\right\rangle_\zeta. \qquad (63)$$

## C.2  Solution for the saddle-point equation

Now our goal is to reduce Eq.(60) for a function of super-vectors to simpler equations for functions of commuting variables. We use representation

$$\psi = \left(\begin{array}{cccc} S_R & \chi_R & S_A & \chi_A^* \end{array}\right)^T, \quad \psi^\dagger = \left(\begin{array}{cccc} S_R^* & \chi_R^* & S_A^* & -\chi_A \end{array}\right) \qquad (64)$$

where $\frac{S_R}{S_R'} = \frac{|S_R|}{|S_R'|}e^{i\theta_R}$ and $\frac{S_A}{S_A'} = \frac{|S_A|}{|S_A'|}e^{i\theta_A}$, and we expand functions of super-vectors over Grassmanian variables $\chi_R, \chi_A, \chi_R^*, \chi_A^*$. It appears to be convenient to look for the solution as function of the arguments $\psi_R^2$ and $\psi_A^2$ and thus to introduce a new function $\tilde{g}_\omega\left(\psi_R^2, \psi_A^2\right) = g_\omega\left(\psi^\dagger\psi, \psi^\dagger\hat{L}\psi\right)$. The expansion of an arbitrary function $f\left(\psi_R^2, \psi_A^2\right)$ over its Grassmanian components looks as follows:

$$f\left(\psi_R^2, \psi_A^2\right) = f\left(|S_R|^2, |S_A|^2\right) + \chi_R^*\chi_R \frac{\partial f\left(|S_R|^2, |S_A|^2\right)}{\partial\left[|S_R|^2\right]} +$$

$$\chi_A^*\chi_A \frac{\partial f\left(|S_R|^2, |S_A|^2\right)}{\partial\left[|S_A|^2\right]} + \chi_R^*\chi_R\chi_A^*\chi_A \frac{\partial^2 f\left(|S_R|^2, |S_A|^2\right)}{\partial\left[|S_R|^2\right]\partial\left[|S_A|^2\right]} \quad (65)$$

To solve Eq.(60) one will need the last term of the above equation only. In these new coordinates, $\psi^\dagger\hat{L}\psi'\psi'^\dagger\hat{L}\psi$ reads as follows:

$$\psi^\dagger\hat{L}\psi'\psi'^\dagger\hat{L}\psi \stackrel{\chi_{R,A}=0}{=} |S_R|^2\left|S_R'\right|^2 + |S_A|^2\left|S_A'\right|^2 - 2\left|S_R\right|\left|S_R'\right|\left|S_A\right|\left|S_A'\right|\cos\left(\theta_R - \theta_A\right) \geq 0 \quad (66)$$

After integration over Grassmanian variables, Eq.(60) acquires the form

$$\tilde{g}_\omega\left(\left|S_R'\right|^2, \left|S_A'\right|^2\right) = \frac{\sigma^\mu N^{1-\gamma}}{\Gamma\left(\frac{\mu}{2}+1\right)} \times \int_0^\infty d\left|S_A\right|^2 d\left|S_R\right|^2$$

$$\int_0^{2\pi}\frac{d\theta}{2\pi}\left[|S_R|^2\left|S_R'\right|^2 + |S_A|^2\left|S_A'\right|^2 - 2\left|S_R\right|\left|S_R'\right|\left|S_A\right|\left|S_A'\right|\cos\theta\right]^{\frac{\mu}{2}} \times$$

$$\frac{\partial^2}{\partial\left[|S_R|^2\right]\partial\left[|S_A|^2\right]}\left\langle e^{i\left(E-\zeta+\frac{\Omega}{2}\right)|S_R|^2 - i\left(E-\zeta-\frac{\Omega}{2}\right)|S_A|^2 - g_\omega\left(|S_R|^2, |S_A|^2\right)}\right\rangle_\zeta \quad (67)$$

In principle, $\tilde{g}_0\left(\left|S_R'\right|^2, \left|S_A'\right|^2\right)$ follows from $\tilde{g}_\omega\left(\left|S_R'\right|^2, \left|S_A'\right|^2\right)$. In this case one should remember the definition $\omega + i0 \equiv \Omega$, so that if $\omega = 0$ than $\Omega = i0$ to maintain the convergence in (60). For our purpose the function $\tilde{g}_\omega\left(\left|S_R'\right|^2, \left|S_R'\right|^2\right)$ is needed (it corresponds to $g_\omega(\psi^+\psi, 0)$ in previous notations). From this point one needs to proceed with the analytical continuation assuming that $\mu > 2$ to obtain reasonable results. It can be calculated in a few steps:

1. Let us define the following function (in order to shorten few next equations):

$$F\left(|S_R|^2, |S_A|^2\right) = \int_0^{2\pi}\frac{d\theta}{2\pi}\left[|S_R|^2 + |S_A|^2 - 2\left|S_R\right|\left|S_A\right|\cos\theta\right]^{\frac{\mu}{2}} \quad (68)$$

with the property

$$\frac{\partial^2 F\left(|S_R|^2, |S_A|^2\right)}{\partial\left[|S_R|^2\right]\partial\left[|S_A|^2\right]}\Bigg|_{|S_R|^2=|S_A|^2} = \frac{\left[|S_R|^2\right]^{\frac{\mu}{2}-2}}{\sqrt{\pi}}\frac{2^\mu}{4}\frac{\mu}{2}\frac{\Gamma\left(\frac{\mu-1}{2}\right)}{\Gamma\left(\frac{\mu}{2}-1\right)} \quad (69)$$

and then integrate Eq.(67) by parts:

$$\tilde{g}_\omega\left(|S_R'|^2, |S_R'|^2\right) = \frac{\sigma^\mu N^{1-\gamma}}{\Gamma\left(\frac{\mu}{2}+1\right)} \left[|S_R'|^2\right]^{\frac{\mu}{2}} \times$$

$$\left\{ \int_0^\infty d\,|S_A|^2\, \frac{\partial F}{\partial\left[|S_A|^2\right]}\bigg|_{|S_R|^2=0} \left\langle e^{-i\left(E-\zeta-\frac{\Omega}{2}\right)|S_A|^2 - \tilde{g}_\omega\left(0, |S_A|^2\right)} \right\rangle_\zeta + \right.$$

$$+ \int_0^\infty d\,|S_R|^2\, \frac{\partial F}{\partial\left[|S_R|^2\right]}\bigg|_{|\phi_A|^2=0} \left\langle e^{i\left(E-\zeta+\frac{\Omega}{2}\right)|S_R|^2 - \tilde{g}_\omega\left(|S_R|^2, 0\right)} \right\rangle_\zeta +$$

$$\left. + \int_0^\infty d\,|S_A|^2\, d\,|S_R|^2\, \frac{\partial^2 F\left(|S_R|^2, |S_A|^2\right)}{\partial\left[|S_R|^2\right]\partial\left[|S_A|^2\right]} \left\langle e^{i\left(E-\zeta+\frac{\Omega}{2}\right)|S_R|^2 - i\left(E-\zeta-\frac{\Omega}{2}\right)|S_A|^2 - \tilde{g}_\omega\left(|S_R|^2, |S_A|^2\right)} \right\rangle_\zeta \right\}$$

$$(70)$$

2. In case of smooth distribution one can use the following trick

$$\left\langle e^{i(E-\zeta)\left(|S_R|^2 - |S_A|^2\right)} \right\rangle_\zeta \approx 2\pi P_D(E)\, \delta\left(|S_R|^2 - |S_A|^2\right) \tag{71}$$

so that (70) reduces to

$$\tilde{g}_\omega\left(|S_R'|^2, |S_R'|^2\right) = \frac{2^{\mu-1}\sigma^\mu N^{1-\gamma} P_D(E)}{\Gamma\left(\frac{\mu}{2}\right)\Gamma\left(\frac{\mu}{2}-1\right)} \Gamma\left(\frac{\mu-1}{2}\right) \left[|S_R'|^2\right]^{\frac{\mu}{2}} \sqrt{\pi} \times$$

$$\int_0^\infty d\left[|S_R|^2\right] \left[|S_R|^2\right]^{\frac{\mu}{2}-2} e^{i\Omega|S_R|^2 - \tilde{g}_\omega\left(|S_R|^2, |S_R|^2\right)} \tag{72}$$

3. Using the fact that $\tilde{g}_\omega\left(|S_R'|^2, |S_R'|^2\right) = \left[|S_R'|^2\, \Gamma_\omega\right]^{\mu/2}$, one reduces the integral equation to the form of transcendental equation

$$\Gamma_\omega = \left[ \frac{2^{\mu-1}\sigma^\mu N^{1-\gamma} P_D(E)}{\Gamma\left(\frac{\mu}{2}\right)\Gamma\left(\frac{\mu}{2}-1\right)} \Gamma\left(\frac{\mu-1}{2}\right) \sqrt{\pi} \int_0^\infty dx\, x^{\frac{\mu}{2}-2} e^{i\Omega x - [x\Gamma_\omega]^{\mu/2}} \right]^{\frac{2}{\mu}} =$$

$$= \left[ \frac{2^{\mu-1}\sigma^\mu N^{1-\gamma} P_D(E)}{\Gamma\left(\frac{\mu}{2}\right)} \Gamma\left(\frac{\mu-1}{2}\right) \sqrt{\pi} \int_0^\infty dr\, L_{\frac{\mu}{2}}(r)\, [-i\Omega + r\Gamma_\omega]^{1-\frac{\mu}{2}} \right]^{\frac{2}{\mu}} \tag{73}$$

which solves the saddle point equation (60) for any $\omega$. In particular, in the limit $\omega \to 0+$ one obtains the result (22).

# D   Large frequencies asymptotics

In this section we derive (35). We need to use Mellin transform defined as

$$\mathcal{M}_f(\lambda) \equiv \int_0^\infty dx\, f(x) x^{\lambda-1} \tag{74}$$

with the property

$$\int_0^\infty dx\, f(x) g(x) = \int_{c-i\infty}^{c+i\infty} \frac{d\lambda}{2\pi i} \mathcal{M}_f(\lambda) \mathcal{M}_g(1-\lambda). \tag{75}$$

$c$ is the constant determined in a way that both $\mathcal{M}_f(\lambda)$ and $\mathcal{M}_g(1-\lambda)$ exist. Applying this to the integral in (33) one receives precise expression

$$\int_0^\infty dr \frac{L_{\mu/2}(r)}{\Omega + 2i\Gamma_\omega r} = \frac{1}{2i\Gamma_\omega} \int_{c-i\infty}^{c+i\infty} \frac{d\lambda}{2\pi i} \frac{2}{\mu} \Gamma\left(\frac{2}{\mu}(1-\lambda)\right) \Gamma(\lambda) \left(-\frac{i\Omega}{2\Gamma_\omega}\right)^{-\lambda}, \quad 0 < c < 1. \quad (76)$$

One can approximate it, counting only the nearest poles contribution $\lambda = 1, 1 + \frac{\mu}{2}$. That gives

$$\int_0^\infty dr \frac{L_{\mu/2}(r)}{\Omega + 2i\Gamma_\omega r} \approx \frac{1}{2i\Gamma_\omega} \left[ \frac{2\Gamma_\omega}{-i\Omega} - \frac{\mu}{2} \left(\frac{2\Gamma_\omega}{-i\Omega}\right)^{\frac{\mu}{2}+1} \right] \quad (77)$$

After substituting this into (33), (25) one will obtain (35) result. The same trick can be used to obtain second order approximations of (21) and (34).

## E   Efetov parameterization

Efetov parametrization for 4-dimensional super-matrix $\hat{Q}$ is defined as follows:

$$\hat{Q} \equiv \hat{T}^{-1}\hat{L}\hat{T} \equiv \begin{pmatrix} \hat{U}_1 & 0 \\ 0 & \hat{U}_2 \end{pmatrix} \hat{\Lambda} \begin{pmatrix} \hat{U}_1^{-1} & 0 \\ 0 & \hat{U}_2^{-1} \end{pmatrix}, \quad \hat{\Lambda} = \begin{pmatrix} \lambda_B & 0 & i\mu_B & 0 \\ 0 & \lambda_F & 0 & \mu_F^* \\ i\mu_B^* & 0 & -\lambda_B & 0 \\ 0 & \mu_F & 0 & -\lambda_F \end{pmatrix} \quad (78)$$

Here $\hat{U}_{1,2}$ are Grassmannian matrices defined as

$$\hat{U}_1 = \exp \begin{pmatrix} 0 & -\alpha^* \\ \alpha & 0 \end{pmatrix} = \begin{pmatrix} 1 - \frac{\alpha^*\alpha}{2} & -\alpha^* \\ \alpha & 1 + \frac{\alpha^*\alpha}{2} \end{pmatrix}, \quad \hat{U}_2 = \exp i \begin{pmatrix} 0 & -\beta^* \\ \beta & 0 \end{pmatrix} = \begin{pmatrix} 1 + \frac{\beta^*\beta}{2} & -i\beta^* \\ i\beta & 1 - \frac{\beta^*\beta}{2} \end{pmatrix} \quad (79)$$

$$\hat{U}_1^{-1} \begin{pmatrix} 1 & 0 \\ 0 & -1 \end{pmatrix} \hat{U}_1 = \begin{pmatrix} 1 - 2\alpha^*\alpha & -2\alpha^* \\ -2\alpha & -1 - 2\alpha^*\alpha \end{pmatrix}, \quad \hat{U}_2^{-1} \begin{pmatrix} 1 & 0 \\ 0 & -1 \end{pmatrix} \hat{U}_2 = \begin{pmatrix} 1 + 2\beta^*\beta & -2i\beta^* \\ -2i\beta & -1 + 2\beta^*\beta \end{pmatrix} \quad (80)$$

and $\hat{\Lambda}$ contains the following commuting variables

$$\lambda_B = \cosh\theta_B, \quad \lambda_F = \cos\theta_F, \quad \mu_B = e^{i\phi_B}\sinh\theta_B, \quad \mu_F = e^{i\phi_F}\sin\theta_F, \quad (81)$$

$$\text{Constraints} = \begin{cases} 0 \le \theta_B < \infty, & 1 \le \lambda_B < \infty \\ 0 \le \theta_B \le \pi, & -1 \le \lambda_F \le 1 \\ 0 \le \phi_{B,F} \le 2\pi \end{cases}$$

with the following relations

$$|\mu_B|^2 = \lambda_B^2 - 1, \quad |\mu_F|^2 = 1 - \lambda_F^2 \quad (82)$$

Mesure of integration over Efetov matrix $\hat{Q}$ reads as

$$d\hat{Q} = -\frac{d\lambda_B d\lambda_F d\phi_B d\phi_F}{(\lambda_B - \lambda_F)^2} d\alpha d\alpha^* d\beta d\beta^* \quad (83)$$

# F Evaluation of the integral in Eq.(42)

Is this section we provide details on how we obtained the result shown in Eq.(43). The starting point is the integral in (42). Since the large parameter is $\kappa = \frac{\pi\omega}{\Delta} \gg 1$, one needs to obtain an answer up to the first order in $1/\kappa \ll 1$. If $\omega \sim E_{th}$ then $\kappa\frac{\omega}{\Gamma} \sim 1$ so that it is reasonable to denote $\frac{\omega}{\Gamma}$ as $\frac{p}{\kappa}$, $\quad p \sim 1$. With these notations integral in (42) will take the form

$$\mathcal{Y} = \mathcal{Y}_1 + \frac{p}{\kappa^2}\mathcal{Y}_2 \tag{84}$$

$$\mathcal{Y}_1 = \frac{1}{2}\int_1^\infty d\lambda_B \int_{-1}^1 d\lambda_F \left(1 + 2i\frac{p}{\kappa}\lambda_B\right)^2 e^{i\kappa(\lambda_B - \lambda_F) - p\lambda_B(\lambda_B - \lambda_F)} = \tag{85}$$

$$\frac{1}{2i\kappa}\int_1^\infty d\lambda_B \frac{\left(1 + 2i\frac{p}{\kappa}\lambda_B\right)^2}{1 + i\frac{p}{\kappa}\lambda_B} e^{i\kappa(\lambda_B - 1)\left(1 + i\frac{p}{\kappa}\lambda_B\right)} \left(e^{2i\kappa\left(1 + i\frac{p}{\kappa}\lambda_B\right)} - 1\right)$$

$$\mathcal{Y}_2 = \int_1^\infty d\lambda_B \int_{-1}^1 d\lambda_F \frac{\lambda_B}{\lambda_B - \lambda_F} e^{i\kappa(\lambda_B - \lambda_F) - p\lambda_B(\lambda_B - \lambda_F)} \tag{86}$$

$\mathcal{Y}_1$ is easily integrated over $\lambda_F$, while $\mathcal{Y}_2$ requires an additional step. Let us use

$$\frac{d\mathcal{Y}_2}{dp} = -\int_1^\infty d\lambda_B \int_{-1}^1 d\lambda_F \lambda_B^2 e^{i\kappa(\lambda_B - \lambda_F) - p\lambda_B(\lambda_B - \lambda_F)} = \tag{87}$$

$$\frac{i}{\kappa}\int_1^\infty d\lambda_B \frac{\lambda_B^2}{1 + i\frac{p}{\kappa}\lambda_B} e^{i\kappa(\lambda_B - 1)\left(1 + i\frac{p}{\kappa}\lambda_B\right)} \left(e^{2i\kappa\left(1 + i\frac{p}{\kappa}\lambda_B\right)} - 1\right)$$

Both integrals collects on the $\lambda_B - 1 < \frac{1}{\kappa}$ scale so that one can make $\lambda_B = 1 + \frac{x}{\kappa}$ substitution and then expand over $\frac{1}{\kappa}$ parameter up to the lowest order. Note that $\Omega \equiv \omega + i0$ maintains the convergence.

$$\mathcal{Y}_1 \approx \int_0^\infty dx \frac{ie^{(i-0)x}}{2\kappa^2} \left(1 - e^{2i\kappa - 2p}\right) = \frac{1}{2\kappa^2}\left(e^{2i\kappa - 2p} - 1\right) \tag{88}$$

$$\frac{d\mathcal{Y}_2}{dp} = \frac{i}{\kappa^2}\int_0^\infty dx e^{(i-0)x}\left(e^{2i\kappa - 2p} - 1\right) = \frac{1}{\kappa^2}\left(1 - e^{2i\kappa - 2p}\right) \Rightarrow \mathcal{Y}_2 = \frac{2p + e^{2i\kappa - 2p}}{2\kappa^2} + \text{const} \tag{89}$$

To restore the constant we apply $p = 0$. This integral is easily evaluated after its derivation over $\kappa$ and later integration. Constant can be found in $\kappa \to \infty$ limit.

$$\mathcal{Y}_2\bigg|_{p=0} = \frac{i}{\kappa} + \frac{e^{2i\kappa} - 1}{2\kappa^2} \Rightarrow \mathcal{Y}_2 = \frac{e^{2i\kappa - 2p} - 1 + 2p}{2\kappa^2} + \frac{i}{\kappa}. \tag{90}$$

As one can see from (84), the second term lowest order is much smaller so it is enough to consider $\mathcal{Y} \approx \mathcal{Y}_1$ only. Having restored all the notations, one should obtain the final expression (43).

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
