# Peer review of "Density of states correlations in Lévy Rosenzweig-Porter model via supersymmetry approach"

_SciPost Physics_

## Round 2 · Referee Report · Anonymous (Referee 1) · 2025-7-25

Report

The paper provides a fundamental contribution to the analysis of spectral correlations in Rosenzweig-Porter
random matrix model, with heavy-tailed distribution of matrix entries. The model is important as it
provides a (relatively) simple framework for studying the intriguing phase with unusual Nonergodic Extended (NEE) states,
believed to be relevant for description of interacting quantum manybody systems with disorder.
Although some information of level correlations in such a model has been available already in the regime of large energy separation,
computing it explicitly for lower energy regime represented a considerable technical challenge.
The authors managed to adapt a version of the Efetov supersymmetry approach used previously for analysis of sparse random matrices
to the present model. Such an adaptation and especially the subsequent analysis of the saddle-point solution and
ensuing level correlations in various regimes looks highly nontrivial, even for those familiar with supersymmetry computations.
It certainly required new analytical insights reflecting physics of the problem.
The most profound one is the additive Ansatz used for the solution of the saddle-point equation, eq.(19)-(20), which
reflects statistical independence of real and imaginary parts of the self-energy, observation going back to Ref.3.
The authors provide enough technical detail and overall logic for their computations, though it is not very easy to follow
without really trying to repeat them at technical level. Probably it is inevitable for papers of such nature. Nevertheless those strongly motivated to understand the
technical details seemenly should be able to follow the presentation. Summary of the results is highly useful.
I therefore strongly recommend this highly non-trivial and rich paper for publication in your journal.

Remark: a seemingly important technical trick is presented in eq.(71). It is written that such a formula is justified for "smooth distributions".
It would be helpful to explain more explicitly what means "smooth" in this context: what is the scale which ensures validity of this approximation?

Recommendation

Publish (surpasses expectations and criteria for this Journal; among top 10%)

---

## Round 2 · Referee Report · Anonymous (Referee 2) · 2025-9-12

Disclosure of Generative AI use

The referee discloses that the following generative AI tools have been used in the preparation of this report:

In preparing this referee report, I used OpenAI’s ChatGPT (GPT-5) to assist with:

Formatting the report according to SciPost Physics guidelines (summary, strengths, weaknesses, detailed comments, conclusion).

Improving the clarity and structure of the language, while keeping the scientific content and judgments fully my own.

Suggesting phrasing for constructive feedback to the authors.

All scientific assessments, critical remarks, and final recommendations in this report are mine. The generative AI tool was used only as a drafting and language-structuring aid, not as a source of scientific analysis.

Strengths

Strengths

The topic is timely and relevant for the communities studying Anderson localization, non-ergodic delocalized phases, and many-body localization.

The analytical derivation via supersymmetry is highly non-trivial and constitutes a rare exact treatment of such an ensemble.

The results potentially connect to fundamental questions about wave packet dynamics, multifractality, and phase transitions in disordered systems.

Weaknesses

Weaknesses

The manuscript is presented in a very technical manner, with many steps left unexplained for non-specialists.

Important background references are missing, especially on correlations and related models.

Several definitions (e.g. IPR) and motivations (e.g. why certain techniques fail, why supersymmetry is needed) are not clearly provided.

Figures are few, difficult to read, and poorly interpreted.

The physical meaning and implications of the results are not sufficiently discussed.

Report

General summary

The manuscript presents an exact derivation of the correlations of the density of states for a random matrix ensemble recently introduced to describe the emergence of a non-ergodic delocalized phase in disordered quantum systems, such as many-body localization or the Anderson transition in infinite effective dimension. This ensemble, the Lévy–Rosenzweig–Porter model, extends the classical Rosenzweig–Porter model by including broadly distributed off-diagonal elements with power-law tails.

The study of density-of-states (DOS) correlations is important since it provides information about system dynamics, such as the return probability of a wave packet. It also usually encodes multifractality signatures (in particular the dimension $D_2$) and characterizes characteristic time scales. Exact analytical approaches are scarce in this field, and the treatment of broad distributions is highly non-trivial.

Overall, this is an interesting and technically strong work, but in its current form the manuscript suffers from presentation issues, lack of pedagogical explanations, and missing contextual references.

Conclusion and recommendation

The manuscript addresses an interesting problem with a rare exact approach. However, in its present form the presentation is overly technical and difficult to follow, with missing references, insufficient physical interpretation, and unclear figures.

I believe the work has potential to be publishable in SciPost Physics, but substantial revisions are required before publication. I therefore recommend Major Revision.

Requested changes

Introduction

The statement “almost no exact theoretical results are available” is too strong; there exist some mathematical results that should be cited.

The discussion of “correlations” is unclear. Several works (Mirlin et al., Roy et al.) on the role of correlations should be cited. In addition, it is not obvious why this aspect is emphasized here since correlations of disorder are not included in the present model.

The text often assumes a very strong background from the reader. For example, the sentence “however, to study level correlations at not-so-large energies a more elaborated technique is needed” should be clarified: why does the cavity method fail in this regime?

Definitions and methodology

Some basic quantities, such as the Inverse Participation Ratio (IPR), are not defined.

In Sec. 2.2, references should be added to works explaining in more detail the supersymmetric techniques employed.

The notation of supervectors overlaps with that of eigenstates, which makes the presentation confusing.

In Sec. 3, the authors should explain why the functional integral approach is needed here, and why a standard Hubbard–Stratonovich decoupling cannot be applied.

Derivations and results

The derivations are presented in a very technical way. While the appendix gives details, the main text should highlight more clearly which aspects are non-trivial, and what physical insights are gained only thanks to this method.

The predictions should be more deeply discussed: what do they imply for return probability, multifractality (is there a signature of $D_2$?), and comparisons with Gaussian RP, log-normal RP, or cavity methods?

Possible connections to Anderson or MBL transitions should be commented on.

Figures

Figures are rare, hardly legible, and their nature is unclear (are these numerical simulations or analytic curves?).

They should be made more reproducible, better captioned, and more extensively interpreted.

Recommendation

Ask for major revision

  • validity: high
  • significance: good
  • originality: good
  • clarity: ok
  • formatting: reasonable
  • grammar: reasonable

Author:  Mikhail Feigel'man  on 2025-09-15  [id 5816]

(in reply to Report 2 on 2025-09-12)

I am grateful to the Referee for his report finally arrived. Some of the comments will definitely be used by the authors to increase our quality of presentation. However, some other comments are rather cryptic: which exactly references the Referee wants us to include ? It is not at all evident from the Report. Please clarify this point in order to speed up appropriate action of the authors and also in order to avoid further delays with manuscript handling.

M. Feigel'man

Anonymous on 2025-09-26  [id 5864]

(in reply to Mikhail Feigel'man on 2025-09-15 [id 5816])
Disclosure of Generative AI use

The comment author discloses that the following generative AI tools have been used in the preparation of this comment:

correction of english formulation.

Category:
pointer to related literature

The references I have in mind include those by Imbrie and De Roeck, Huveneers et al. on mathematically rigorous results concerning MBL, as well as several papers by Roy and Logan, and the work of Mirlin et al. [Phys. Rev. B 109, 214203 (2024)] on the effects of correlations on the Anderson transition in random graphs.
I believe the authors are already familiar with the key references on the supersymmetry approach in quantum disordered systems. What would be particularly helpful for a non-expert reader, however, is guidance on where to find additional background on known results invoked in the paper—for example, a pointer to Sec. II of [the relevant book] …

---

## Round 2 · Referee Report · Anonymous (Referee 3) · 2025-9-24

Disclosure of Generative AI use

The referee discloses that the following generative AI tools have been used in the preparation of this report:

Polish the text, make it more fluent, and correct the grammar.

Strengths

1- The authors provide a detailed analytical calculation of the two-point density–density correlation function, a central and technically demanding quantity;

2- The topic is timely and relevant, as it connects with ongoing interest in fractal and nonergodic phases in disordered and complex systems;

Weaknesses

1- The analysis is extremely technical and dense, which may limit accessibility to a broader physics audience not already familiar with the methods of random matrix theory;

2- The surprising coincidence with the Gaussian Hermitian RP ensemble (Eq. (43)) is mentioned but not explained, leaving a gap in the physical understanding of the result;

3- The manuscript contains a few typos and typographical mistakes, which detract from the overall readability and polish of the paper;

Report

The authors study the Hermitian Lévy-Rosenzweig–Porter ensemble. A central result of the paper is the detailed analytical calculation, through the supersymmetric approach, of the two-point density-density correlation function, which they show exhibits nontrivial behavior depending on the energy scale considered. The work extends previous studies of Hermitian RP ensembles and provides new insights into the structure of non-Hermitian random matrix ensembles with tunable delocalization properties.

I found the results compelling and the paper timely. The calculations are highly nontrivial and technically challenging, clearly reflecting significant effort and expertise. Despite the technical nature of the analysis, the authors have made some effort to guide the reader through the derivations, providing useful physical interpretations and connecting the mathematical results to the underlying physics. However, some derivations remain difficult to follow, which may limit accessibility to a broader physics audience not already familiar with the supersymmetric approach. Nevertheless, both the subject matter and methodology are intrinsically highly technical.

I have one scientific question that I would like the authors to address. If I understand correctly, Eq. (43), which describes the behavior of the two-point function on energy scales much larger than the mean level spacing but much smaller than the Thouless energy, coincides with the corresponding result for the Gaussian Hermitian RP ensemble. This seems rather surprising, especially given the differences in the microscopic structure of the local spectrum between the two models, as discussed in the text. I would encourage the authors to comment further on this point and, if possible, provide a physical explanation for this apparent convergence in behavior despite the underlying structural differences. Is this similarity is expected from general theoretical principles or represents a surprising coincidence?

Finally, the manuscript contains a few typos, misprints, and typographical errors. I recommend that the authors carefully proofread and revise the text before publication.

Overall, I consider this paper an important and valuable contribution to the study of random matrix ensembles, and I recommend it for publication once the above points are addressed.

Requested changes

1- Clarification of Eq. (43) similarity to Gaussian Hermitian RP ensemble: Provide additional commentary on why the two-point function behavior on intermediate energy scales (larger than mean level spacing, smaller than Thouless energy) coincides with the Gaussian Hermitian RP ensemble result, and possibly offer a physical explanation for this apparent convergence despite the differences in microscopic spectral structure between the Lévy and Gaussian RP models;

2- Consider adding more explanatory text or intuitive descriptions for the most technical derivations; Ensure that physical motivations are clearly stated before diving into technical calculations;

3- Conduct thorough proofreading to eliminate typos, misprints, formatting, and typographical errors throughout the text;

Recommendation

Ask for minor revision

---

## Editorial Decision

resubmitted